# Post-Training Quantization via Residual Truncation and Zero Suppression for Diffusion Models

## Abstract

Diffusion models achieve high-quality image generation but face deployment challenges due to their high computational requirements. Although 8-bit outlier-aware Post-Training Quantization (PTQ) matches full-precision performance, extending PTQ to 4 bits remains challenging. Larger step sizes in 4-bit quantization amplify rounding errors in dense, low-magnitude activations, leading to the loss of fine-grained textures. We hypothesize that not only outliers but also small activations are critical for texture fidelity. To this end, we propose Quantization via Residual Truncation and Zero Suppression (QuaRTZ), a 4-bit PTQ scheme for diffusion models. QuaRTZ applies 8-bit min–max quantization for outlier handling and compresses to 4 bits via leading-zero suppression to retain LSBs, thereby preserving texture details. Our approach reduces rounding errors and improves quantization efficiency by balancing outlier preservation and LSB precision. Both theoretical derivations and empirical evaluations demonstrate the generalizability of QuaRTZ across diverse activation distributions. Notably, 4-bit QuaRTZ achieves an FID of 6.98 on FLUX.1-schnell, outperforming SVDQuant that requires auxiliary FP16 branches.

## 1 Introduction

Diffusion models have emerged as the state-of-the-art in generative modeling, achieving remarkable performance in text-to-image synthesis, super-resolution, and inpainting (Ho et al., 2020; Rombach et al., 2022; Podell et al., 2023; Peebles & Xie, 2023; Black-Forest-Labs, 2024). However, the iterative refinement process is computationally expensive, which limits its deployment in latency- and resource-constrained environments such as mobile devices, on-device AI assistants, or large-scale cloud serving with strict throughput demands.

Low-bit quantization reduces memory footprint, bandwidth demand, and arithmetic cost, while enabling efficient execution on modern accelerators (Han et al., 2015). Post-training quantization (PTQ) is particularly attractive for diffusion models where fine-tuning is costly, as it requires neither retraining nor access to the original dataset (Nagel et al., 2020; Li et al., 2021). While 8-bit and even 6-bit PTQ for diffusion models have proven to be effective (Li et al., 2023; Huang et al., 2024; Ryu et al., 2025), pushing to 4-bit precision (W4A4) remains challenging due to error propagation; quantization noise accumulates over hundreds of timesteps, degrading image texture quality.

Existing PTQ methods address this issue by focusing on outlier preservation through temporal alignment (Huang et al., 2024; He et al., 2023; Chen et al., 2024b) or condition-aware scaling (Ryu et al., 2025; Li et al., 2023); however, this overlooks the dominant error source at extremely low precision — the loss of Least Significant Bits (LSBs). Diffusion models refine subtle variations over many iterations (Ho et al., 2020; Peebles & Xie, 2023), making them especially sensitive to rounding errors near zero. With activations densely concentrated around small values, truncating LSBs discards critical fine-grained information and leads to collapsed generations. Tackling the conflicting challenge of preserving both outliers and LSBs is essential to making 4-bit quantization practical for diffusion models.

To this end, we propose Quantization via Residual Truncation and Zero suppression (QuaRTZ), a novel two-stage 4-bit quantization scheme for diffusion models. The first stage minimizes rounding error through 8-bit min-max uniform quantization, resulting in a fine-grained integer representation of the original value. The second stage compresses integer representations to a targeted 4 bits using a Leading Zero Suppression (LZS) kernel, which preserves a salient 4-bit representation beginning from the top-most activated bit. This process allows for high entropy of the compressed representation, where the magnitude of the outliers is retained and LSBs are preserved without information loss. Our two-stage design simultaneously protects outliers and LSBs, directly addressing the two dominant sources of error at low precision.

Our contributions are as follows:

- We propose QuaRTZ, a novel 4-bit quantization scheme that successfully balances outlier preservation and LSB precision.

- Our QuaRTZ scheme demonstrates state-of-the-art performance in various diffusion architectures, including UNet and DiT backbones. Notably, our W4A4-quantized model outperforms SVDQuant counterparts even without an error compensation module, enabling 3.8x reduction in memory footprint compared to the 16-bit baseline.

- We illustrate the effectiveness of QuaRTZ in theoretical, information, empirical and hardware perspectives in depth, providing core reasoning behind the insight of preserving LSBs.

## 2 RELATED WORKS

Diffusion models have established state-of-the-art performance in a wide range of image generation tasks, including unconditional generation (Ho et al., 2020; Rombach et al., 2022) and text-to-image synthesis (Rombach et al., 2022; Podell et al., 2023; Sauer et al., 2024; Chen et al., 2024a; Black-Forest-Labs, 2024). Recent extensions integrate transformer backbones, further scaling model capacity and controllability (Peebles & Xie, 2023; Chen et al., 2024a; Black-Forest-Labs, 2024). Despite these advances, the inherently iterative denoising process results in slow inference, posing a significant barrier to deployment in latency- or resource-constrained environments.

Quantization has emerged as a promising direction to accelerate diffusion models by reducing memory footprint and enabling efficient low-precision arithmetic. Two main paradigms exist: Quantization-Aware Training (QAT), which jointly learns task objectives and quantization parameters (Esser et al., 2019), and Post-Training Quantization (PTQ), which applies quantization to pre-trained models without retraining. While QAT generally achieves higher accuracy at low bit-widths, it requires complete training data and considerable computational resources. PTQ, in contrast, does not require complete data or finetuning, making it a practical path for scaling large generative models where retraining is often infeasible.

Consequently, the main focus of recent PTQ research has been primarily on developing methods for handling outliers. Approaches such as PTQ4DM (Shang et al., 2023), Q-Diffusion (Li et al., 2023), and TFQM-DM (Huang et al., 2024) mitigate large-magnitude errors through temporal alignment, calibration strategies, or condition-aware scaling (Li et al., 2023; He et al., 2023; Chen et al., 2024c). Building on these foundations, DGQ (Ryu et al., 2025) achieved W4A6 quantization for text-to-image models, and SVDQuant (Li et al., 2024b) reached 4-bit precision by introducing 16-bit LoRA (Hu et al., 2022) branches to absorb quantization error. However, prior work largely fails to maintain image quality below 6-bit precision. While SVDQuant is successful at maintaining image quality, it limits the efficiency gains of full low-bit quantization because it relies on auxiliary FP16 branches. These branches introduce additional parameters, modify the original architecture, and require mixed-precision fusion, which increases implementation complexity and reduces deployment efficiency.

In this work, we propose a 4-bit post-training quantization method for diffusion models that preserves fine-grained texture quality while improving efficiency without auxiliary, higher-precision branches. We focus on both outliers and LSBs simultaneously, departing from prior methods that emphasize only outlier preservation.

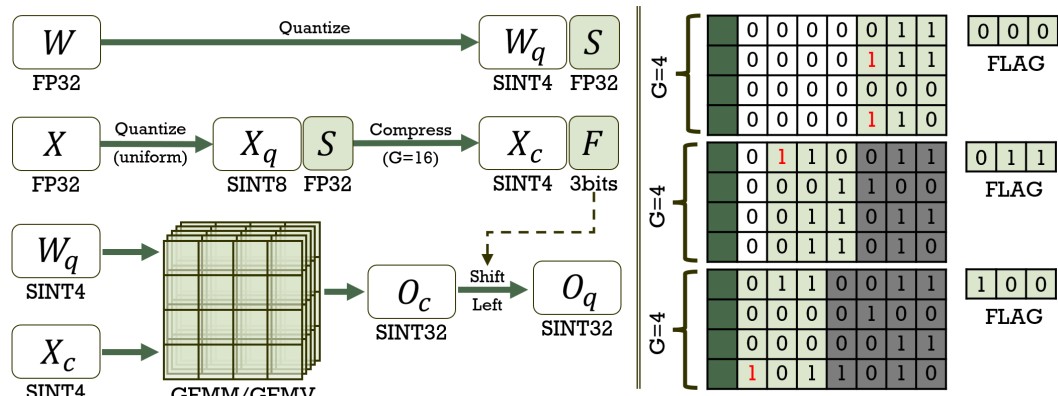

Figure 1: Illustration of the proposed two-stage quantization. (Left) We show 4-bit inference pipeline using QuaRTZ INT4. Weights are quantized to `SINT4` directly. For activation, we first apply uniform 8-bit integer quantization to capture outliers with a relatively small step size, and then compress activations to 4 bits via subgroup-based leading-zero suppression. 4-bit representations are multiplied and accumulated, followed by `left-shift` operation using `FLAG`. (Right) We illustrate examples scenarios of compression using LZS. This process allows to preserve small, high density activations remain unchanged while outliers maintain their magnitude without significant information loss. In our experiments, we use group size 16 to perform sub-group compression.

# 3 QUANTIZATION VIA RESIDUAL TRUNCATION AND ZERO SUPPRESSION

We hypothesize that both outliers and LSBs are crucial in diffusion models. Outliers drive large corrections and the generation of salient features, while LSBs capture fine variations that shape textures and smooth gradients. Our two-stage quantization scheme preserves both outliers and LSBs with minimal information loss.

We target outliers and LSBs in two stages: 8-bit quantization and 4-bit quantization as illustrated in Figure 1. In the first stage, we apply 8-bit integer quantization, that captures the sparse outlier distribution while keeping rounding error small due to the fine step size. The 8-bit representation spans the full dynamic range, leaving relatively few outliers and concentrating most information in the LSBs.

Let $x$ and $\hat{x}$ denote the original and quantized activations, respectively. The quantization process is defined as:

$$\hat{x} = \text{clamp}\left(\left\lfloor \frac{x}{s} \right\rceil + z, ; -127, ; 127\right), \quad s = \frac{x_{\max} - x_{\min}}{255}, \quad z = \left\lfloor -\frac{x_{\min}}{s} \right\rceil, \tag{1}$$

where $s$ and $z$ denote the scaling factor and zero point, and $x_{\min}, x_{\max}$ are the minimum and maximum activation values.

In the second stage, we exploit the redundancy of 8-bit codes using Leading Zero Suppression (LZS) to compress them into 4 bits. The key idea is to discard unused high-order zeros while preserving both the magnitude of outliers and the precision of LSBs. Each 8-bit signed integer is reformatted into a 1-bit sign $sgn$ and a 7-bit magnitude $mag$, and values are grouped into $K$ blocks of size $G_s$ (e.g., 16 or 32 elements). Within each block, we compute a shared flag that indicates the most significant active bit. Specifically, the flag is derived from the number of leading zeros, computed via the CUDA intrinsic `clz` function on the bitwise OR of all magnitudes in the subgroup (NVIDIA Corporation, 2025) as:

$$\text{FLAG} = \max\left(29 - \texttt{clz}(m), 0\right), \quad \text{FLAG} \in \{0, 1, 2, 3, 4\} \tag{2}$$

where $m$ denotes the aggregated magnitude. The counting leading zero (`clz`) function counts consecutive high-order zero bits in a 32-bit integer, which returns a value from 0 to 32. Thus, we subtract 3 from 32 to preserve the bottom 3 bits when $mag$ is smaller than 8.

A `right-shift` operation of FLAG bits is then applied to all values in the block, yielding a compact signed 4-bit representation. This process retains salient bits and suppresses redundant high-order zeros, achieving compression with minimal information loss. During inference, since each group has been shifted equally, the output of Matrix Multiply-Accumulate (MMA) can be adjusted by the FLAG bit `left-shift` operation.

## 4 ANALYSIS OF QUARTZ

We analyze QuaRTZ from multiple perspectives, including theoretical distortion bounds, bit-wise entropy, empirical evaluations, and latency.

### 4.1 DISTORTION ANALYSIS

We show that our two-stage quantization scheme provides a lower upper bound on quantization error compared to direct 4-bit min-max uniform quantization. Detailed derivation of our inequality is described in Appendix A.

**Theorem 1** (Error Bound for QuaRTZ). *Let $X \in \mathbb{R}$ with density $p(x)$. Denote the quantization error of direct 4-bit uniform quantization as $E_q^4$, and the error of 8-bit quantization followed by LZS compression as $E_{total}$. If less than half of the probability mass lies in high-index bins ($|j| \geq 8$), then*

$$E_{total} < E_q^4. \tag{3}$$

*Sketch of Proof.* For uniform $n$-bit quantization, the error is bounded by $E_q^n \leq s_n/2$. With $s_4 = 16s_8$, the sufficient condition becomes $E_{\text{LZS}} < 7.5\,s_8$.

The expected LZS error is

$$\mathbb{E}[E_{\text{LZS}}] = s_8 \sum_{k=4}^{7} P_k(2^{k-3} - 1), \tag{4}$$

where $P_k = \mathbb{P}(H = k)$ denotes the density of $k$-th bit in $X$. In the worst case, where every value with 4 truncated bits, $E_{\text{LZS}} \leq 15s_8 \cdot \mathbb{P}(|J| \geq 8)$. If $\mathbb{P}(|J| \geq 8) < 0.5$, then the condition is satisfied. $\square$

### 4.2 BIT-WISE ENTROPY ANALYSIS

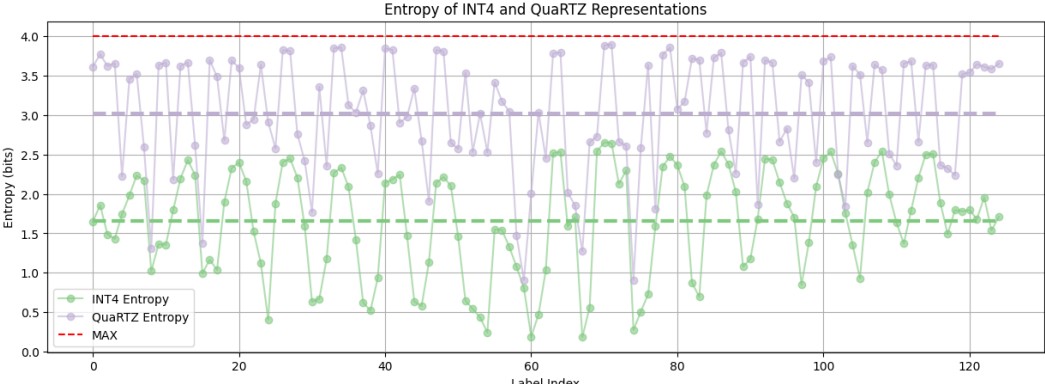

Figure 2: Entropy analysis demonstrates that our method exhibits higher entropy at every layer compared to naïve INT4.

We present the entropy of INT4 and QuaRTZ representations at each layer in Figure 2. For every layer, our method has higher entropy compared to the INT4 min-max uniform quantization method.

Higher entropy indicates that all four bits are activated with nearly equal frequency. The results show that QuaRTZ constructs compact and informative 4-bit representations by removing the redundancy of 8-bit codes. Although increasing representation entropy was not an explicit design goal, this improvement is a direct result of our primary motivation of exploiting the redundancies of 8-bit values.

## 4.3 EMPIRICAL ANALYSIS

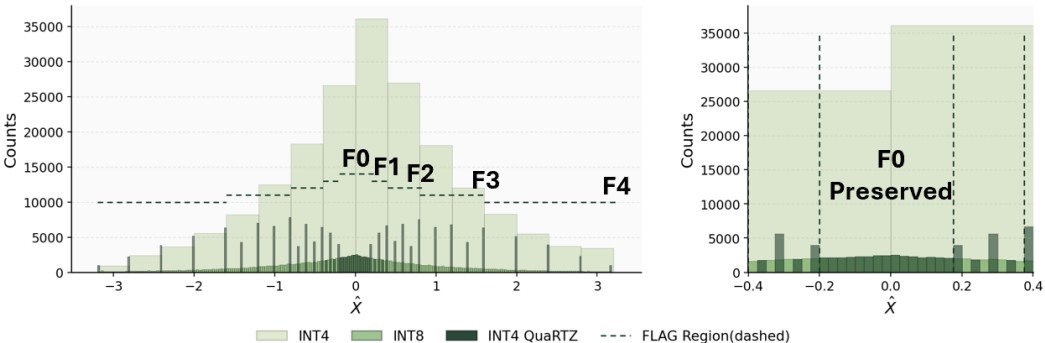

Figure 3: Compared to naïve INT4 quantization, QuaRTZ avoids severe rounding errors in dense low-magnitude regions. The histogram is partitioned into FLAG regions (F0–F4): F0 denotes the preserved fine-grained region around zero, while F1–F4 correspond to progressively larger magnitude ranges captured via FLAG-based shifts. Despite compression, the magnitude of outliers is retained similarly to INT4 quantization.

Effectiveness of QuaRTZ is further supported by empirical analysis on random values, as shown in Figure 4. The results demonstrate that our method preserves LSBs effectively compared to direct 4-bit integer quantization, which suffers from severe rounding errors. Additionally, the magnitude of outliers is well retained in both cases, supporting our claim that LSBs also play a critical role in generating high-quality images.

## 4.4 LATENCY ANALYSIS

We show that our method is computationally efficient and hardware-friendly with a comprehensive analysis of the 4-bit QuaRTZ kernel on various layer size in Table 1. On a RTX 4090 GPU with native s4 Tensor Core MMA, GEMM executes as $s4 \times s4 \to s32$, and the per-group power-of-two scale is applied as an integer left shift on the $s32$ accumulators inside the $K$-loop, adding only $\sim 1/G_s$ extra integer ops per slice with no additional global memory traffic—typically negligible relative to MMA throughput. Meanwhile, activation (A-side) traffic is nearly halved versus int8: activations are stored as packed $s4$, and the only overhead is a single flag byte per subgroup (i.e., $1/G_s$ bytes per element), which stays cache-resident. We also report the latency of various attention settings, power, and area of the proposed kernel in Appendix D.

Table 1: Compression and decompression overhead using LZS kernel ($G = 16$) on RTX 4090 GPU. We show that extra overhead due to shifting is minimal and can be efficiently implemented.

| M | K | Compression (ms) | Decompression (ms) | Overhead (ms) |
|---|---|---|---|---|
| 512 | 3072 | 0.0111 | 0.0042 | 0.0153 |
| 512 | 12288 | 0.0408 | 0.0272 | 0.0680 |
| 4096 | 3072 | 0.0782 | 0.0544 | 0.1326 |
| 4096 | 12288 | 0.3035 | 0.2150 | 0.5185 |
| 4608 | 3072 | 0.0782 | 0.0538 | 0.1320 |
| 4608 | 12288 | 0.4250 | 0.2650 | 0.6900 |
| 4608 | 15360 | 0.3400 | 0.2227 | 0.5627 |

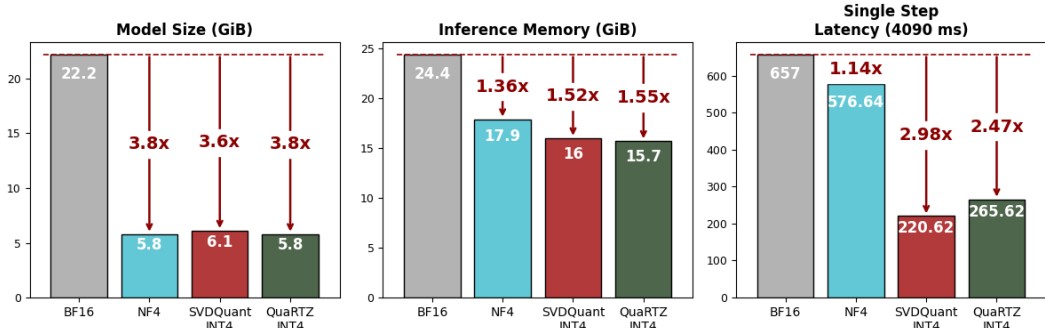

Figure 4: Compared model size, inference memory, and single step latency with different quantization settings on RTX 4090 GPU. We show that our method shows best inference memory efficiency and our quantization kernel is highly competitive compared to SVDQuant.

## 5 EXPERIMENTS AND ANALYSIS

### 5.1 SETUP

**Model and Dataset**  We evaluate our proposed scheme using UNet-based architectures, including LDM (Rombach et al., 2022), Stable Diffusion (SD) v1.4 (Rombach et al., 2022), SDXL-Turbo (Sauer et al., 2024), and DiT-based architectures, such as PixArt-$\Sigma$ (Chen et al., 2024a) and FLUX.1-schnell (Black-Forest-Labs, 2024). Experiments are conducted on widely used LSUN-Bedrooms, LSUN-Churches (Yu et al., 2015), CelebA-HQ (Karras et al., 2018), FFHQ (Karras et al., 2019), MS-COCO (Lin et al., 2014), MJHQ-30K (Li et al., 2024a), and summarized Densely CAptioned Images (sDCI) dataset (Urbanek et al., 2024).

**Quantization Setup**  We follow prior works (Li et al., 2023; Huang et al., 2024; Li et al., 2024b) for calibration and quantization settings for comparison. For unconditional image generation, we sample 256 samples per timestep while 128 prompts are sampled from COCO Captions 2017 (Lin et al., 2014) for text-to-image generation. Generalization performance is evaluated using 5K randomly sampled prompts from the MJHQ-30K and sDCI dataset. Additional details are included in C.

**Metrics**  We assess model performance using Fréchet Inception Distance (FID) (Heusel et al., 2017), CLIP Score (Hessel et al., 2021), and ImageReward (IR) (Xu et al., 2023). LDM models are evaluated with FID, while FID, CLIP Score, and IR are used for other models. We also use LPIPS (Zhang et al., 2018) and Peak Signal Noise Ratio (PSNR) to measure perceptual similarity and numerical similarity of DiT-based models. Results from prior literature are either taken directly from original papers or reproduced under comparable conditions. We generate 30K samples for evaluating LDM models, while 5K samples are used for the rest. All experiments are conducted on a single A100 GPU using PyTorch.

**Baselines**  We compare our work with prior state-of-the-art quantization techniques with TFMQ-DM (Huang et al., 2024), DGQ (Ryu et al., 2025), and SVDQuant (Li et al., 2024b).

The notation W$x$A$y$ indicates that $x$ bits and $y$ bits are used for weight and activation quantization, respectively. Additional experimental details are provided in Appendix F.

### 5.2 MAIN RESULTS

**Unconditional Image Generation**  We first evaluate our method on unconditional image generation using LDM and report the results in Table 2. With the W4A4G16 setting, our method achieves substantial quality improvements over the baseline, narrowing the gap with the W4A8 settings by small margins. We can observe that the direct quantization from 32-bit precision to 4-bit using TFMQ-DM leads to a significant degradation in generation quality across all cases.

Table 2: FID scores of unconditional image generation using LDM-4 on LSUN-Bedrooms 256 × 256, FFHQ 256 × 256, and CelebA-HQ 256 × 256, and LDM-8 on LSUN-Churches 256 × 256. [†] indicates scores from running open-source codes.

| Methods | Bits (W$x$A$y$) | LSUN-Beds | LSUN-Churches | CelebA-HQ | FFHQ |
|---|---|---|---|---|---|
| Full Prec. | W32A32 | 3.47 | 4.34 | 20.54 | 9.67 |
| TFMQ-DM[†] | W4A8 | 6.2 | 13.94 | 21.39 | 10.34 |
| | W4A4 | 327.01 | 327.40 | 224.41 | 275.63 |
| QuaRTZ (Ours) | W4A4 | 7.11 | 14.81 | 23.53 | 14.71 |

Table 3: Quantization results for UNet backbone diffusion model on text-to-image generation task with 4-bit quantization.

| Model | Methods | Bits (W$x$A$y$) | MS-COCO | | |
|---|---|---|---|---|---|
| | | | FID ↓ | CLIP ↑ | IR ↑ |
| SDv1.4 | Full Prec. | W32A32 | 25.03 | 0.265 | 0.189 |
| | TFMQ-DM | W4A6 | 230 | 0.127 | - |
| | DGQ | W4A6 | 43.66 | 0.263 | - |
| | QuaRTZ (Ours) | **W4A4** | **37.64** | **0.264** | 0.065 |
| SDXL-Turbo | Full Prec. | W32A32 | 30.74 | 0.265 | 0.850 |
| | TFMQ-DM | W4A6 | 270.00 | 0.022 | - |
| | DGQ | W4A6 | 45.00 | 0.245 | - |
| | SVDQuant | W4A4 | **24.60** | - | 0.816 |
| | QuaRTZ (Ours) | W4A4 | 30.86 | **0.265** | **0.833** |

Table 4: Quantization results for DiT backbone diffusion model on text-to-image generation task with 4-bit quantization.

| Model | Methods | Bits (W$x$A$y$) | MJHQ | | | | sDCI | | | |
|---|---|---|---|---|---|---|---|---|---|---|
| | | | FID↓ | IR ↑ | LPIPS ↓ | PSNR ↑ | FID↓ | IR ↑ | LPIPS ↓ | PSNR ↑ |
| PixArt-Σ | Full Prec. | W16A16 | 16.61 | 0.953 | - | - | 24.88 | 0.963 | - | - |
| | Naïve INT4[†] | W4A4 | 206.33 | -1.24 | 0.762 | 9.08 | 229.00 | -1.28 | 0.761 | 8.71 |
| | Naïve MX4[†] | W4A4 | 194.30 | -1.44 | 0.746 | 12.71 | 221.58 | -1.65 | 0.776 | 11.85 |
| | Naïve NVFP4[†] | W4A4 | 33.89 | 0.666 | 0.517 | 14.84 | 33.18 | 0.666 | 0.556 | 13.85 |
| | SVDQuant[†] | W4A4 | **16.1** | **0.875** | **0.321** | **17.61** | **16.74** | **0.91** | **0.354** | **16.37** |
| | QuaRTZ (Ours) | W4A4 | 27.89 | 0.841 | 0.460 | 15.08 | 27.51 | 0.873 | 0.492 | 14.02 |
| FLUX.1-schnell | Full Prec. | W16A16 | 19.2 | 0.966 | - | - | 20.88 | 0.974 | - | - |
| | Naïve INT4[†] | W4A4 | 9.13 | **0.963** | 0.345 | 16.31 | 8.51 | 0.988 | 0.353 | 15.27 |
| | Naïve MX4[†] | W4A4 | 9.40 | 0.962 | 0.341 | 16.28 | 8.58 | **0.995** | 0.352 | 15.17 |
| | Naïve NVFP4[†] | W4A4 | 7.23 | 0.955 | 0.280 | 17.52 | 6.90 | 0.994 | 0.284 | 16.39 |
| | SVDQuant[†] | W4A4 | 7.07 | 0.958 | 0.257 | 18.25 | 6.67 | 0.976 | **0.26** | **17.19** |
| | QuaRTZ (Ours) | W4A4 | **6.98** | 0.962 | **0.254** | **18.27** | **6.56** | 0.987 | 0.258 | 17.16 |

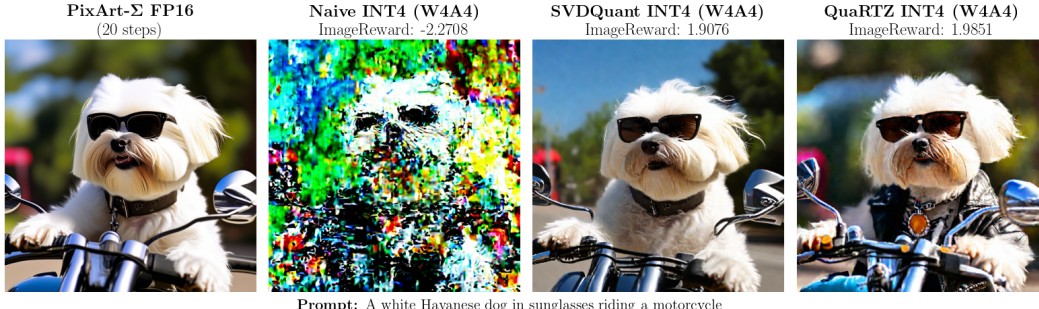

(a) Generated images from different quantization methods on FLUX.1-schnell model on MJHQ dataset.

(b) Generated images from different quantization methods on PixArt-Σ model on MJHQ dataset.

Figure 5: Qualitative comparison on DiT based architectures using different quantization setting.

**Text-to-Image Generation**    We report quantitative results for 4-bit quantization across several diffusion architectures—SDv1.4, SDXL-Turbo, PixArt-Σ, and FLUX.1-schnell—in Table 2 and Table 4. The number of inference steps is set to 50, 4, 25, and 4, respectively. For UNet-based architectures, our method consistently surpasses W4A6 baselines across all metrics, despite operating at lower precision. Notably, it slightly outperforms SVDQuant on FLUX.1-schnell without requiring auxiliary high-precision branches. Compared to naïve INT4 quantization, the proposed two-stage scheme that preserves LSBs yields a substantial accuracy improvement.

On SDXL-Turbo and PixArt-Σ, however, our method shows notable degradation. We attribute this to the error-compensation module in SVDQuant, which explicitly addresses outlier precision, whereas our design prioritizes maintaining LSB fidelity alongside coarse outlier magnitude. This suggests that combining our approach with targeted outlier compensation, such as QwT (Fu et al., 2025) or SVDQuant, may further improve robustness in a low-bit quantization scheme.

### 5.3   ABLATION STUDY

Group size controls the granularity of the LZS operation, creating a trade-off between representation precision and inference latency. We evaluate this effect under the W4A4 setting on the LDM-4 model with the LSUN-Bedrooms dataset. As shown in Figure 6(a), FID score increases approximately linearly as group size increases. This behavior is expected since larger groups increase the likelihood of outliers, which in turn forces truncation in the LSB region. Nevertheless, the visual quality remains stable, suggesting that group size can be tuned according to deployment requirements without significant perceptual degradation. Following the result of Table 8, we recommend using group size of 16 or 32 where latency and image quality are well balanced.

### 5.4   POTENTIAL APPLICATIONS TO LLMS

We further explore the applicability of QuaRTZ to Large Language Models (LLMs). Here, FP16 activations are dynamically quantized and compressed to 4-bit with a group size of $G_s = 8$ using LZS, following the same precedure as Diffusion Models. Weights are quantized directly to

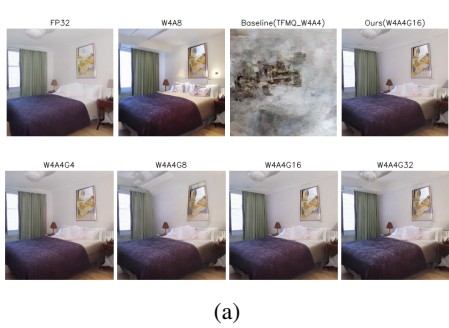
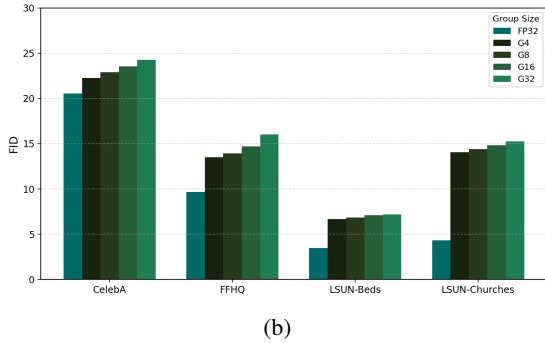

(a)

(b)

Figure 6: (a) (up) Qualitative comparison with baseline models with LDM-4 on the LSUN-Bedrooms dataset. (down) Generated results using different group sizes using QuaRTZ W4A4 with LDM-4 on the LSUN-Bedrooms dataset. (b) FID scores with different group size using LDM-4 model quantized using QuaRTZ.

Table 5: Perplexity comparison between FP16 and QuaRTZ-quantized models. Lower is better.

| Model | #Params | FP16 ↓ | QuaRTZ 4-bit ↓ | $\Delta$ | Relative $\Delta$ |
|-------|---------|--------|----------------|----------|-------------------|
| Qwen2 | 0.5B | 12.35 | 13.67 | +1.32 | +10.7% |
| Qwen2 | 1.5B | 8.87 | 9.37 | +0.50 | +5.6% |
| Qwen2 | 7B | 6.67 | 6.98 | +0.30 | +4.5% |
| LLaMA2 | 7B | 5.12 | 5.35 | +0.23 | +4.5% |
| LLaMA3 | 8B | 5.75 | 6.30 | +0.55 | +9.5% |

SINT4 using GPTQ with group size 128. We use sequence length of 2048. As shown in Table 5, QuaRTZ-quantized models closely follow their FP16 baselines across scales. The increase in perplexity remains modest, with relative error between $+4.5\%$ and $+10.7\%$. These preliminary results suggest that LSB preservation is also beneficial for autoregressive transformers, and demonstrate the potential of QuaRTZ as a general low-bit quantization scheme beyond diffusion models.

## 6 CONCLUSION

This paper introduces QuaRTZ, a novel two-stage PTQ framework that achieves successful 4-bit quantization of diffusion models. We argue that preserving LSBs is as important as capturing outliers. Our method addresses both challenges by applying a two-stage quantization-then-suppression approach, minimizing rounding errors for LSBs while retaining outlier magnitudes. Our theoretical analysis indicates that our method outperforms conventional 4-bit quantization, particularly under distributions with high LSB density such as Gaussian and Laplacian. This theoretical advantage is corroborated by extensive empirical evaluations, which demonstrate superior performance across a variety of diffusion models and tasks. Notably, our method achieves an FID of 6.98 in a W4A4 setting for the FLUX.1-schnell model, surpassing the state-of-the-art W4A4 model.

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

## A  QUANTIZATION ERROR UPPER BOUND DERIVATION

**Setup.** Let $X \in \mathbb{R}$ with PDF $p(x)$. Consider symmetric uniform (midtread) $n$-bit quantizers with step $s_n$ and reconstruction levels $\{q_k\}$ that cover the same dynamic range for $n = 4, 8$. Then

$$E_q^n = \sum_k \int_{\mathcal{B}_k} p(x) \, |x - q_k| \, dx \le \frac{s_n}{2}, \qquad E_q^4 \le \frac{s_4}{2}, \quad E_q^8 \le \frac{s_8}{2}.$$

Since the ranges match, $s_4 = 16 \, s_8$ (7 magnitude bits at 8-bit vs. 3 at 4-bit).

**Signed 8-bit reformat and LZS.** Quantize $X$ to signed int8:

$$x_q = \text{sgn}(x) \cdot m \, s_8, \qquad m \in \{0, 1, \dots, 127\}.$$

We represent each code as [sign] [7-bit magnitude]. LZS keeps the sign bit and *only the top 3 magnitude bits*. Define the magnitude bit-length

$$H(m) = \begin{cases} 0, & m = 0, \\ \lfloor \log_2 m \rfloor + 1, & m \ge 1, \end{cases} \qquad H(m) \in \{0, 1, \dots, 7\}.$$

If $H(m) \leq 3$, all magnitude bits are retained and no truncation occurs. If $H(m) \geq 4$, the lower $H(m) - 3$ magnitude bits are discarded. With truncation toward zero, the additional magnitude error (in LSB units of the 8-bit grid) is bounded by

$$E_{\text{LZS}}(m) = \begin{cases} 0, & H(m) \leq 3, \\ \left(2^{H(m)-3} - 1\right) s_8, & H(m) \geq 4, \end{cases}$$

and the sign is preserved, so there is no sign error.

**Total error bound and dominance condition.** Let $E_{\text{total}}$ be the total error of the signed-LZS path (int8 quantization plus LZS truncation). By triangle inequality,

$$E_{\text{total}} \leq E_q^8 + \mathbb{E}[E_{\text{LZS}}].$$

A sufficient condition for the signed-LZS path to beat naïve signed 4-bit is

$$E_{\text{total}} < E_q^4 \iff \mathbb{E}[E_{\text{LZS}}] < E_q^4 - E_q^8 \leq \frac{s_4 - s_8}{2} = \frac{16 s_8 - s_8}{2} = 7.5 \, s_8.$$

**Expected LZS error under the signed magnitude distribution.** Let $M \in \{0, \ldots, 127\}$ be the magnitude index from signed 8-bit quantization, and $H = H(M)$. Define $P_k = \mathbb{P}(H = k)$ for $k \in \{0, \ldots, 7\}$. Then

$$\mathbb{E}[E_{\text{LZS}}] = s_8 \sum_{k=4}^{7} P_k \left(2^{k-3} - 1\right).$$

Therefore a sufficient condition is

$$\sum_{k=4}^{7} P_k \left(2^{k-3} - 1\right) < 7.5.$$

Equivalently, in terms of bins of the 8-bit *magnitude* quantizer, note that $H(m) \geq 4$ iff $m \geq 8$. Writing

$$P_k = \sum_{m: \, H(m)=k} \int_{x \in \text{bin}(m)} \left(p(x) + p(-x)\right) dx,$$

we get the explicit bound

$$\sum_{m=8}^{127} \left(2^{H(m)-3} - 1\right) \int_{x \in \text{bin}(m)} \left(p(x) + p(-x)\right) dx \; < \; 7.5.$$

**Worst case reduction.** In the worst case all mass with $m \geq 8$ sits at $H(m) = 7$ (4 truncated bits), so $2^{H-3} - 1 = 15$ and

$$15 \sum_{m=8}^{127} \int_{x \in \text{bin}(m)} \left(p(x) + p(-x)\right) dx \; \leq \; 7.5 \quad \Rightarrow \quad \int_{\{|x_q| \geq 8 \, s_8\}} p(x) \, dx \; < \; \tfrac{1}{2}.$$

That is, if less than half of the probability mass is quantized into magnitude indices $m \geq 8$ (i.e., values whose 7-bit magnitudes require $\geq 3$ bits), then signed-LZS satisfies $\mathbb{E}[E_{\text{LZS}}] < 7.5 \, s_8$ and the total error is guaranteed below naïve signed 4-bit's upper bound.

## B    ENTROPY OF 4-BIT REPRESENTATION

We compared the entropy of 4-bit representations of activations at each layer in Figure 8 and Figure 7. For every layer, our QuaRTZ has higher entropy compared to INT4 min-max uniform quantization method. Higher entropy indicates that all four bits are activated with near-equal frequency, thus better utilizing 4 bits to store information.

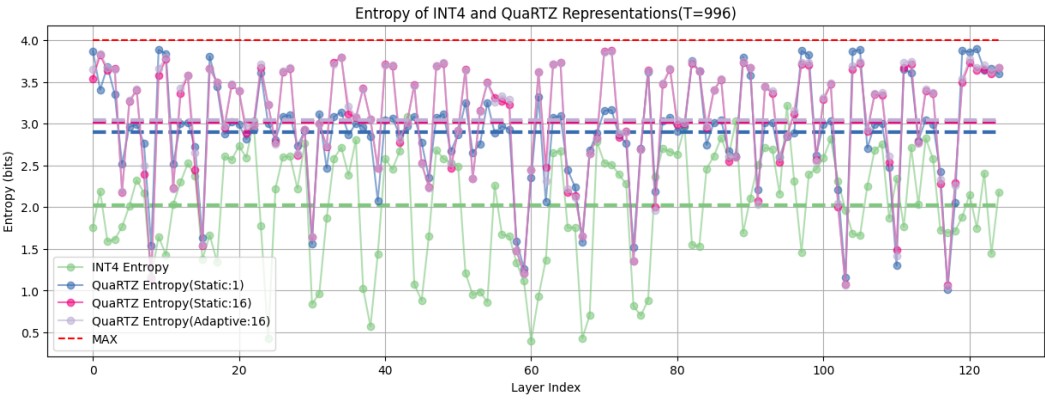

(a) Entropy analysis at timestep 996.

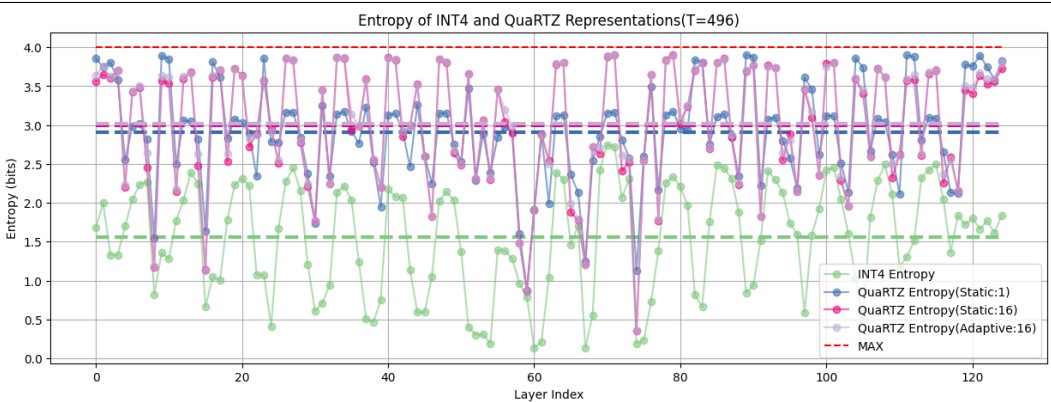

(b) Entropy analysis at timestep 496.

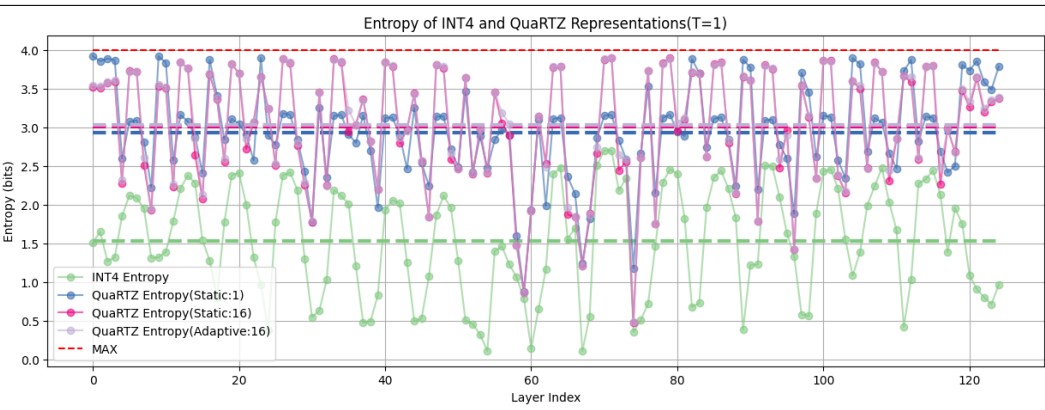

(c) Entropy analysis at timestep 1.

Figure 7: Visualization of 4-bit entropy of quantized values using naïve INT4 min-max uniform quantization and our QuaRTZ method on LDM4 trained on LSUN-Bedrooms averaged at given timestep.

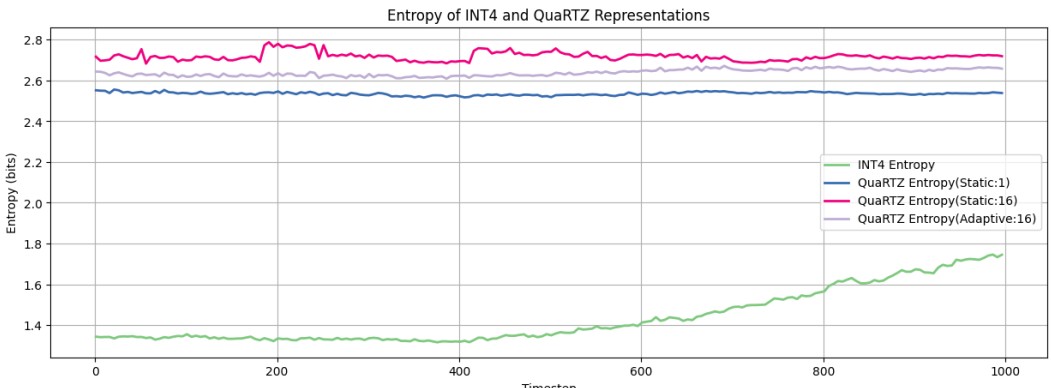

Figure 8: Visualization of 4-bit entropy of quantized values using naïve INT4 min-max uniform quantization and our QuaRTZ method on LDM4 trained on LSUN-Bedrooms averaged over all layers.

## C  EXPERIMENTAL DETAILS

For LDMs, we use per-channel weight quantization and static per-tensor activation quantization. To create 8-bit representation, we kept consistent to TFMQ-DM (Huang et al., 2024) for fair comparison regarding layer selection. Once we acquire the 8-bit representation, 4-bit compression is applied on-the-fly. For SDv1.4, SDXL-Turbo, PixArt-$\Sigma$, and FLUX.1-schnell, we follow the setting with SVDQuant(Li et al., 2024b). Weights and activations are quantized groupwise with a size of 64 with 16-bit scales, then GPTQ is applied to the weights. We note that we do not use smoothing or auxiliary error compensation module.

## D  HARDWARE EFFICIENCY OF QUARTZ KERNEL

Table 6: Comparison of power and area for MAC units.

|  | FP $16 \times 16$ MAC | INT $16 \times 8$ MAC | INT $8 \times 8$ MAC | INT $4 \times 4$ Proposed |
|---|---|---|---|---|
| **Area** ($\mu m^2$) | | | | |
| Multiplier | 3042.2 | 1052.2 | 559.4 | 112 |
| Shifter | 0 | 0 | 0 | 156.5 |
| Reg. + Accm. | 1127.1 | 631 | 431 | 385.3 |
| Total | 4169.3 | 1683.2 | 990.4 | 653.8 |
| **Power** ($mW$) | | | | |
| Multiplier | 0.3378 | 0.0506 | 0.023 | 0.0028 |
| Shifter | 0 | 0 | 0 | 0.0067 |
| Reg. + Accm. | 0.1242 | 0.0733 | 0.0581 | 0.0451 |
| Total | 0.4620 | 0.1239 | 0.0811 | 0.0546 |

## LLM USAGE

We used an AI-based assistant (ChatGPT) solely for minor language editing and polishing. All research ideas, experimental design, and analyses were conducted by the authors.

Table 7: Latency comparison of 4-bit QuaRTZ CUDA kernel and Python implementation across various attention settings. We use A6000 GPU and PyTorch library for Python implementation.

| heads×dim | Group | Python (ms) | QuaRTZ (ms) |
|---|---|---|---|
| 32×128 | g8 | 0.653 | 0.105 |
|  | g16 | 0.531 | 0.102 |
|  | g32 | 0.525 | 0.092 |
| 40×128 | g8 | 0.805 | 0.103 |
|  | g16 | 0.555 | 0.104 |
|  | g32 | 0.502 | 0.096 |
| 64×128 | g8 | 0.749 | 0.107 |
|  | g16 | 0.515 | 0.102 |
|  | g32 | 0.519 | 0.097 |

Table 8: Latency comparison of 4-bit QuaRTZ CUDA kernel and Python implementation across linear layers. We use A6000 GPU and PyTorch library for Python implementation.

| Layer size | Group | Python (ms) | QuaRTZ (ms) |
|---|---|---|---|
| 4096×4096 | g8 | 5.410 | 0.189 |
|  | g16 | 5.057 | 0.184 |
|  | g32 | 5.017 | 0.176 |
| 5120×5120 | g8 | 7.678 | 0.383 |
|  | g16 | 6.965 | 0.268 |
|  | g32 | 6.389 | 0.239 |
| 8192×8192 | g8 | 18.74 | 0.468 |
|  | g16 | 16.50 | 0.327 |
|  | g32 | 15.01 | 0.313 |

