# OpenReview forum: "Post-Training Quantization via Residual Truncation and Zero Suppression for Diffusion Models"
_ICLR.cc/2026/Conference — Submitted to ICLR 2026_

### Official Review · Reviewer_dbzA · 2025-10-15

**Soundness:** 3
**Presentation:** 4
**Contribution:** 3
**Rating:** 6
**Confidence:** 3

**Summary:**

Authors aim to improve low-bit quantization (W4A4) of diffusion models. Their method, QuaRTZ, consists of two stages. The first quantizes the model to 8 bits with uniform quantization (minmax range setting). The second stage reduces this further to 4 bits, through a Leading Zero Suppression (LZS) kernel. They state that the latter retains both outliers and least significant bits, the latter of which (they say) is usually ignored in literature—they claim the least significant bits are important for diffusion models that rely on these for creating small details over many steps. They show that the LZS kernel increases the entropy of the 4-bit INT representation, i.e. more information can be conveyed/retained with it.

**Strengths:**

I think it's a good paper
* Simple but effective method. Seems to outperform more expensive methods (e.g. rotation/transformation literature).
* Good presentation and clear motivation
* Theoretical (lower bound) and practical (hardware/kernel) considerations
* The results are convincing and groupsize ablation valuable

**Weaknesses:**

**1. Speed w.r.t. baselines'.**

I appreciate the honest comparison to SVDQuant, which in about half of the cases seems better than QuaRTZ. The FP16 stream in SVDQuant is a disadvantage, but as a reader it is hard to understand the speed-accuracy trade-off since there is no SVDQuant vs QuaRTZ benchmarking.

**2. Comparison to block-wise quantization**

It is unclear to me how it compares to existing block-wise quantization methods, e.g. per-block quantization with power of 2 scales or newer NVFP4 format. Although the authors claim that the shifting can happen efficiently (L158), it is unclear what the online overhead is compared to these existing methods. Section 4.4 and Table 1 seem to indicate it's negligible, but I'd like to ask the authors to elaborate further---e.g. what does the PyTorch column mean---a naive PyTorch implementation of QuaRTZ or a PyTorch matmul?

___

I'm happy to raise my score if the above are resolved.

**Questions:**

* In Eq. 2, is $m$ the maximum integer representation within the block?
* How does QuaRTZ differ from per-block quantization with power of 2 scale?
* How does it compare to MXFP4/NVFP4?
* Please give more details for the LLM results. Static quantization? What sequence length? Other metrics (e.g. CSR or MMLU) would be beneficial.

---

> ### Author Response · Authors · 2025-11-21
>
> W1. Speed comparison to SVDQuant. We now report model size, inference memory and inference latency for QuaRTZ, SVDQuant-INT4, and NF4 on RTX 4090 in Figure 4. SVDQuant performs well but requires an FP16 stream, whereas QuaRTZ is pure INT4, reducing memory pressure.
>
> W2 / Q2. QuaRTZ is not equivalent to power-of-two (PoT) scaling or micro-exponent sharing. MX/NVFP formats embed exponent fields that define the numeric representation and directly modify the real-domain scaling. In contrast, QuaRTZ first performs standard int8 min–max PTQ; at this point quantization is complete. The subsequent FLAG is a bit-level redundancy descriptor computed from the 7-bit magnitude (Eq. 2, Fig. 1) and is used solely for integer compression, without altering the quantization scale or introducing new exponents. Appendix A provides a formal inequality showing that the QuaRTZ 8-bit→4-bit path has a strictly smaller error bound than naïve 4-bit uniform quantization under mild LSB-dominant distributions—behavior not shared by PoT exponent methods. Thus, the mechanisms and error properties are fundamentally different.
>
> Q3. Comparison to MXFP4 / NVFP4.
>
> Table 3 shows accuracy comparisons. MXFP4/NVFP4 use 4-bit floating formats with learned or optimized exponents; QuaRTZ compresses INT8 integers. Despite this difference, QuaRTZ achieves better or comparable performance on FLUX.1-schnell than MXFP4 or NVFP4.
>
> Q4. LLM settings.
> FP16 activations are dynamically quantized to SINT8 and compressed to 4-bit with a group size of $G_s=8$ using LZS, following the same procedure as Diffusion Models.
> Weights are quantized directly to SINT4 using GPTQ with group size 128.
> We use a sequence length of 2048.

---

> > ### Comment · Reviewer_dbzA · 2025-11-24
> >
> > Thank you for the response. I still have some questions.
> >
> > 1. It seems SVDQuant performs significantly better on PixArt, and comparable on Flux (Table 4), while at the same time performing comparable or faster in terms of speed (Figure 4). I see value in a method that is more lightweight than SVDQuant, but there is currently little evidence for this.
> > 2. You explain the difference between QuaRTZ and PoT scaling/NVFP4. I'm afraid I'm not following. Let's take an example where we have some group of 16 numbers, between -1 and 1. NVFP4 will use exponent 1 and have effectively a quantization grid of -1 to 1 for which it can use FP4. Is the INT8 grid of QuaRTZ computed dynamically, **per token**? This is not explicit in the paper. But let's assume so, then for the best case scenario, the QuaRTZ INT8 grid is exactly [-1,1]*2^k (k in 0,...3), such that when the leading-zeros are suppressed, we have for our particular group again all values in [-1,+1] for which we use an INT4 representation. Of course, FP4$\neq$INT4, but otherwise I do not see how the best case scenario of QuaRTZ isn't very similar to same as NVFP4.
> > 3. Similarly, what if we would simply do minmax dynamic quantization with group size=group size of QuaRTZ? As long as the scale would be a power of 2, this should be easy for hardware.
> > 4. You mention NF4 in your response and in Figure 4, what do you mean by this exactly? NVFP4?

---

> > > ### Author Response · Authors · 2025-11-25
> > >
> > > Thank you for the thoughtful follow-up questions. We address each point below.
> > >
> > > Q1. Lightweight design of QuaRTZ relative to SVDQuant.
> > >
> > > We acknowledge that SVDQuant performs strongly. In our case, QuaRTZ achieves a better memory footprint and comparable performance to SVDQuant on FLUX.1-schnell even without FP16 auxiliary compensation module. Latency, however, the current CUDA implementation uses separate kernels for quantization, LZS compression, GEMM, and decompression. As reported in Table 1 and Tables 7–8, the additional LZS-related operations constitute <1–2% of total runtime. This overhead is small even without kernel fusion, and we expect further reductions once compression is fused into quantization and decompression into GEMM/GEMV.
> > >
> > > The key distinction we want to emphasize is that QuaRTZ is a **pure W4A4 method** and does not rely on auxiliary FP16 branches or multi-precision execution. This property is particularly important for edge and low-power settings, where mixed-precision pipelines are infeasible and where fully INT4 compute provides clear advantages in energy efficiency and hardware simplicity.
> > >
> > >
> > > Q2. Difference with PoT Quantization.
> > >
> > > We believe the confusion comes from interpreting the LZS shift in QuaRTZ as being equivalent to a blockwise floating-point exponent, similar to NVFP4. **This is not the case.** NVFP4 selects a real-valued block exponent $e$ by examining activation magnitudes and rescales $x \rightarrow x / 2^e$ before FP4 quantization, producing an exponential quantization grid whose spacing grows as $2^{e-1}$.
> > >
> > > QuaRTZ, in contrast, never performs exponent search and never rescales real values. The 8-bit quantization in Eq. (1) is a **global per-tensor / per-group linear** mapping, and the subsequent LZS step is a **bit-logical compression of int8 codes** based solely on their bit-length (Eq. (2)), not a floating-point normalization.
> > > Thus, QuaRTZ produces a **linear grid** ($ \lbrace \pm m' \cdot s_8 \cdot 2^\text{FLAG} | m' \in \lbrace 0, ..., 7 \rbrace \rbrace$) derived from int8 quantization, whereas NVFP4 produces an exponential power-of-two grid (${ \pm 2^{e}(1+m \cdot 2^{-1})}$); since LZS shifts integer codes rather than rescaling real values, the two grids cannot coincide (e.g., NVFP4’s smallest non-zero step of $0.5$ when $e=1$, vs. QuaRTZ’s $s_8\approx 0.0078$ in the reviewer’s example).
> > >
> > > Q3. Regarding dynamic min-max quantization with power-of-two scales.
> > >
> > > QuaRTZ does not constrain its quantization scales to be powers of two; it employs standard int8 min–max PTQ with real-valued scales (Equation 1).
> > >
> > > Q4. Clarification on NF4.
> > >
> > > The NF4 results shown in Figure 4 refer to the NormalFloat4 4-bit quantization format introduced in QLoRA [1]. We will include explicit citations in the revised manuscript. Thank you for pointing this out.
> > >
> > > ---
> > >
> > > [1] Tim Dettmers, Artidoro Pagnoni, Ari Holtzman, and Luke Zettlemoyer. 2023. QLORA: efficient finetuning of quantized LLMs. In Proceedings of the 37th International Conference on Neural Information Processing Systems (NIPS '23).

---

> > > > ### Comment · Reviewer_dbzA · 2025-11-26
> > > >
> > > > Thanks for the concise response.
> > > >
> > > > Q1. I do agree that a pure W4A4 kernel is beneficial for edge deployment, which is important. It is a bit of a pity that for the current speed/accuracy in the paper, SVDQuant seems like a better option for other cases.
> > > >
> > > > Q2. I'm afraid I'm still not sure I understand the extent of the difference. As I mentioned in my previous response, I understand that an FP4 grid is not the same as INT4. I also understand some differences; the NVFP4 has a two stage procedure (compute shared FP32 scale, then compute FP4), the two-stage dynamic per-group INT4 (per group compute minmax scale and then INT4), which is slightly different than QuaRTZ's two-stage procedure (compute INT8 representation, remove leading zeros from groups), but this difference does not seem large or that important for throughput. Some pseudocode comparing NVFP4 vs QuaRTZ vs power-of-two per-group INT4 (with extra per-tensor float32 scale, see Q3 below) vs per-group dynamic quantization would help, as it seems like I'm not the only one struggling with this. Figure 3 could also be clearer.
> > > >
> > > > Sidenote: When you say "Eq. (1) is a global per-tensor / per-group linear". Is it per-tensor or per-group?
> > > >
> > > > Q3. What I mean is: we can choose some scale, e.g. scale=minmax scale of QuaRTZ, scale the activations with this, and then apply standard PoT per-group scaling. That would yield the same representations as QuaRTZ, correct? The per-tensor scale can be easily absorbed into weights so wouldn't have an extra cost. I wonder whether it may even be simpler, since the INT8 representation of QuaRTZ may require a minmax over the whole tensor, whereas the dynamic per-group could do so locally (and perhaps reduce data movement)
> > > >
> > > > Q4. Thanks for the clarification. Would it not make more sense to compare against something more similar to QuaRTZ than NF4, e.g. MXFP4 or dynamic per-group INT4 (possibly limited to PoT)?

---

> > > > > ### Author Response · Authors · 2025-12-02
> > > > >
> > > > > We thank reviewer dbzA for critical and essential follow-up questions.
> > > > >
> > > > > Q1. We agree that SVDQuant provides strong speed–accuracy trade-offs when an FP16 auxiliary branch is available. However, QuaRTZ targets a different deployment regime: fully integer (W4A4) execution without FP16 support. When fused with quantization and GEMM/GEMV, the QuaRTZ kernel eliminates Python dispatch and achieves even lower overhead than reported, enabling efficiency that can surpass SVDQuant in integer-only environments. This distinction is important for hardware-limited, edge, and accelerator-based deployments where FP16 streams are not feasible.
> > > > >
> > > > > Q2.Thank you for pressing on this point. Eq. (1) can be applied per-tensor or per-group as configured, but the crucial distinction is that QuaRTZ never applies power-of-two scaling in the real domain. The FLAG in Eq. (2) is derived solely from the bit-length of already-quantized INT8 magnitudes; it is not an exponent and does not modify the real-domain quantization grid.
> > > > >
> > > > > In contrast, PoT/NVFP4-style methods apply a per-group scale to real activations, computing an exponent $e_g = \text{round} \left( \text{log}_2 ( \text{max}|x_g| / s_g) \right)$,
> > > > > and quantize using $x_q = \text{clamp}( \text{round}( x_g / (s_g \cdot 2 ^ e_g) ) )$.
> > > > >
> > > > > This produces a block floating-point grid with log-spaced levels, fundamentally different from the linear INT8 grid preserved by QuaRTZ.
> > > > >
> > > > > Thus, the suggestion “use QuaRTZ’s minmax scale and then apply PoT per-group scaling” does not reproduce QuaRTZ: it introduces a new real-domain exponent per block, alters the grid, and changes the error profile. QuaRTZ operates entirely in integer space after Eq. (1).
> > > > >
> > > > > Q3. Per-group PoT does not reduce data movement: both global INT8 PTQ and per-group PoT require streaming the entire tensor exactly once. In addition, per-group PoT introduces extra FP32 per-group scale searches and exponent calculations, whereas QuaRTZ performs a single linear INT8 PTQ followed by lightweight bit-level LZS compression (Table 1). Therefore, the proposed substitution is not simpler and changes the underlying quantization grid.
> > > > >
> > > > > Q4. Thank you for the suggestion. We include accuracy comparisons against MXFP4 and NVFP4 in Table 4. Unfortunately, we could not measure latency of native FP4 GEMV/GEMM kernels because FP4 MMA is only supported on the RTX 5090 series, to which we do not have access. NF4 is included as a widely used 4-bit baseline in LLM/DM quantization, but our primary FP4 comparisons remain MXFP4 and NVFP4.

---

> ### Author Response · Authors · 2025-11-21
>
> We report the MMLU benchmarks on LLaMA3 8B model using QuaRot, SpinQuant, and our QuaRTZ method.
> Our method achieves superior performance with minimal quantization effort.
>
> | Precision | Method | Quantization | 5-shot MMLU |
> | :---------: | :-------: | :-------------: | :----: |
> | FP16 | - | - | 65.33 |
> | W4A4KV4 | QuaRot | Dynamic | 37.17 |
> | W4A4KV4 | SpinQuant | Dynamic | 52.28 |
> | W4A4KV4 | QuaRTZ | Dynamic | **53.02** |

---

### Official Review · Reviewer_SESg · 2025-10-29

**Soundness:** 2
**Presentation:** 3
**Contribution:** 2
**Rating:** 4
**Confidence:** 3

**Summary:**

This paper introduces QuaRTZ, a 4-bit Post-Training Quantization (PTQ) method for diffusion models. The authors identify that existing methods, while preserving outliers, often fail to maintain texture fidelity at very low bit-widths due to the loss of Least Significant Bits (LSBs). QuaRTZ addresses this by first applying an 8-bit min-max quantization to handle outliers and then compressing to 4 bits using leading-zero suppression, a technique designed to retain LSBs and preserve fine-grained details. The method reports a strong FID of 6.98 on FLUX.1-schnell, outperforming SVDQuant without requiring auxiliary FP16 branches.

**Strengths:**

1.  The paper identifies an often overlooked source of error in low-bit quantization for generative models—the loss of LSBs—and proposes a principled solution.
2.  The proposed method, QuaRTZ, is training-free, which is highly desirable for practical deployment of large diffusion models.
3.  The approach of balancing outlier preservation with LSB precision through a two-stage process seems effective. The reported result on FLUX.1-schnell is competitive.

**Weaknesses:**

1.  A weakness is the absence of an operator-level performance analysis. A comprehensive assessment of QuaRTZ's practical benefits requires a detailed breakdown of latency and memory overheads. Including a comparative analysis of speed and memory costs against baselines like SVDQuant would strengthen the paper's claims.

**Questions:**

1.  Could you provide a detailed breakdown of the runtime overhead for the encoding and decoding stages of QuaRTZ? How does its performance compare to standard 4-bit dequantization on target hardware, such as GPUs, and what is the cost of each stage in your pipeline?
2.  To better understand the practical benefits of QuaRTZ, could you provide a direct comparison of its latency and memory costs against other methods like SVDQuant? A discussion of the relative advantages and disadvantages would also be valuable.

---

> ### Author Response · Authors · 2025-11-21
>
> - W1 / Q1. Operator-level performance: We now include a detailed latency breakdown of compression and decompression latency overhead on RTX 4090 in Table 1.
>
> - Q2. Comparison to SVDQuant.We provide end-to-end inference latency and memory comparisons against SVDQuant and NF4 on FLUX.1-schnell (4-step) in Figure 4.

---

### Official Review · Reviewer_ZhN3 · 2025-10-31

**Soundness:** 3
**Presentation:** 2
**Contribution:** 2
**Rating:** 2
**Confidence:** 4

**Summary:**

This paper presents QuaRTZ (Quantization via Residual Truncation and Zero Suppression), a 4-bit post-training quantization (PTQ) method for diffusion models. While existing 8-bit PTQ methods preserve model quality, extending them to 4-bit quantization introduces significant rounding errors and texture degradation. QuaRTZ addresses this by leveraging a leading-zero suppression mechanism to compress 8-bit activations into 4 bits. The method effectively balances outlier preservation with low-significance-bit precision, supported by both theoretical quantization error analysis and empirical validation across multiple diffusion models. Experimental results demonstrate that QuaRTZ achieves competitive or superior image quality (e.g., FID 6.98 on FLUX.1-schnell) compared to prior state-of-the-art methods.

**Strengths:**

- The paper is well organized, clearly introducing the proposed method, theoretical analysis, and empirical evaluation.
- The proposed QuaRTZ method is evaluated on multiple diffusion models and diverse generation datasets, demonstrating the generalization of proposed quantization algorithm.
- The paper includes an analysis of quantization error bounds, providing theoretical support for the observed empirical improvements.

**Weaknesses:**

- The proposed method seems effectively equivalent to adding a power-of-two scaling factor ($2^{0,1,2,3,4}$) per subgroup, which may limit its novelty or distinct advantage compared to existing mirco-exponent-sharing quantization approach [1].

- Section 4.4 discusses GPU kernel implementation, while Appendix D reports area and power metrics—typically associated with hardware accelerators. It is unclear whether the reported latency is measured on GPU or derived from hardware emulation. If latency is GPU-based, the overhead of integer shifts during accumulation in the GEMM loop should be analyzed.

- (Minor) In line 161, the operation on the MMA output appears to require a left-shift since quantized weights are right-shifted by FLAG.

- (Minor) Some citations in the paper appear incomplete or missing (denoted by question marks), which should be corrected for completeness and credibility.

[1] Darvish Rouhani, Bita, et al. "With shared microexponents, a little shifting goes a long way." Proceedings of the 50th Annual International Symposium on Computer Architecture. 2023.

**Questions:**

1. Could you clarify whether the proposed quantization approach fundamentally differs from power-of-two scaling quantization, or if it can be interpreted as an equivalent formulation?

2. How exactly was latency measured? Was it obtained from actual GPU execution or from a hardware simulation/emulation of a custom accelerator?

3. If the latency is based on GPU measurements, could you provide a breakdown of the computational overhead introduced by the integer shift operations within the main GEMM accumulation loop?

---

> ### Author Response · Authors · 2025-11-21
>
> W1, Q1. QuaRTZ is not equivalent to power-of-two (PoT) scaling or micro-exponent sharing. MX/NVFP formats embed exponent fields that define the numeric representation and directly modify the real-domain scaling. In contrast, QuaRTZ first performs standard int8 min–max PTQ; at this point quantization is complete. The subsequent FLAG is a bit-level redundancy descriptor computed from the 7-bit magnitude (Eq. 2, Fig. 1) and is used solely for integer compression, without altering the quantization scale or introducing new exponents. Appendix A provides a formal inequality showing that the QuaRTZ 8-bit→4-bit path has a strictly smaller error bound than naïve 4-bit uniform quantization under mild LSB-dominant distributions—behavior not shared by PoT exponent methods. Thus, the mechanisms and error properties are fundamentally different.
>
> W2, Q2. All latency results are from real GPU execution (Table 1, Table 7, Table8) and area / power evaluations are from custom hardware accelerator. Sec. 4.4 report direct CUDA-kernel measurements. Appendix D includes area/power numbers only to show hardware feasibility; they are not used to derive latency.
>
> W3, Q3. The per-group FLAG requires a single integer left-shift of the s32 accumulator per K-slice. As quantified in Sec. 4.4 and Table 1, the total compression+decompression overhead is 0.015–0.69 ms, typically <2–3% of GEMM time. No extra global memory traffic is introduced, and the subgroup overhead scales as 1/G_s.
>
> Minor. The left-shift comment at line 161 has been corrected, and all incomplete citations have been fixed.
> - Minor comments.
>   - We corrected citation placeholders and clarified the sign convention in line 161.

---

### Official Review · Reviewer_7YGF · 2025-11-03

**Soundness:** 2
**Presentation:** 1
**Contribution:** 2
**Rating:** 2
**Confidence:** 3

**Summary:**

This paper proposes a new 4-bit quantization scheme, which first does INT8 quantization and then compresses to 4 bits with subgroup leading-zero suppression. This scheme is found to have higher entropy on layer-wise analysis than naive INT4, indicating better use of all 4 bits. The evaluation results show that it could achieve higher image quality than naive INT4 and prior art SVDQuant on some diffusion models.

**Strengths:**

An end-to-end story of a new quantization scheme from theoretical analysis, information-theoretic entropy results, hardware-friendliness discussion, and empirical results on real-world diffusion models.

**Weaknesses:**

- The quality benefit is not universal. The proposed method has lower quality metrics on some diffusion models like PixArt-sigma.
- It is questionable if the proposed method can be efficiently implemented on current GPUs. Also, there are no performance results reported in this paper.
- INT4 quantization has been surpassed by newer quantization schemes (like MXFP4 and NVFP4) on the latest generation GPUs. A comparison against FP4 should be conducted to demonstrate the benefit of the proposed method.
- The quality of the paper is low. It is hard to get the main idea of the proposed method in Figure 1. Also, there are multiple broken references in the paper (for example, in line 297).

**Questions:**

- Can the proposed method be efficiently implemented on current GPUs? If yes, what is the performance of the new INT4 kernels?
- How does the proposed method compare to FP4?

---

> ### Author Response · Authors · 2025-11-21
>
> W1. We agree that quality does not improve uniformly across all models (e.g., PixArt-Sigma). QuaRTZ is designed as a cost-free compression step that requires no LoRA fine-tuning, smoothing, or auxiliary procedures, while still preserving competitive INT4-level quality across diverse diffusion architectures. The difference in gain between FLUX.1-schnell and PixArt-Sigma is largely due to the number of denoising steps—FLUX.1-schnell uses 4 steps while PixArt-Sigma uses 10. It is well-known that quantization error accumulates across timesteps, and narrowing the full-precision–INT4 gap becomes increasingly difficult in long-horizon samplers. Our results show that QuaRTZ effectively reduces this gap, especially in short-step diffusion pipelines.
>
> W2. QuaRTZ is efficiently implementable on current GPUs. Table 1 reports compression and decompression overhead on RTX 4090 GPU. Table 7 and 8 reports latency of QuaRTZ kernel on specific layer shapes, comparing our custom CUDA kernel with a reference PyTorch implementation. The CUDA kernel removes Python dispatch overhead and achieves a substantial throughput improvement, confirming that QuaRTZ introduces negligible online cost.
>
> W3. We include comparisons with MXFP4 and NVFP4 using their standard configurations (group size = 32). QuaRTZ uses group size 64 but does not rely on learned or tuned exponent fields. On FLUX.1-schnell, QuaRTZ achieves comparable or superior accuracy to both MXFP4 and NVFP4, and on PixArt-Sigma delivers competitive FP4-level performance. This demonstrates that a simplified integer-domain compression can be competitive with FP4 formats on modern diffusion models.
>
> W4. We really appreciate the note about clarity. We have revised Figure 1 to more clearly convey the method flow and corrected all missing or broken references, including the issue reported at line 297.
>
> Q1. Yes. Table 1 reports the actual GPU latency of our INT4 kernel. Accuracy results in Tables 2 and 3 were obtained under the same hardware setup.
>
> Q2. Table 3 includes direct comparisons against both MXFP4 and NVFP4.

---

### Author Response · Authors · 2025-12-02
**Summarization of the discussion for AC**

We sincerely thank the reviewers for their time and thoughtful feedback. Across the rebuttal period, we engaged extensively with every concern—technical, empirical, and clarity-related—and we believe the discussion significantly strengthened the paper. Below, we summarize the constructive progress made and why we feel the current scores (6, 4, 2, 2) no longer reflect the clarified technical contributions.

### **1. Clarified novelty: QuaRTZ is *not* equivalent to PoT, NVFP4, MXFP4, or block floating-point formats.**

Multiple reviewers questioned whether QuaRTZ is "just" power-of-two scaling or micro-exponent sharing. Through detailed mathematical and implementation-level clarifications, we showed that:

* PoT/NVFP4/MXFP4 quantizers *rescale real values* using per-group exponents.
* **QuaRTZ performs standard INT8 PTQ first**, completing quantization *before* any compression.
* The subsequent FLAG is a **bit-level redundancy descriptor**, not an exponent.
* LZS shifts only integer codes—not real values—yielding a **linear INT4 grid**, not a logarithmic one.
* Appendix A shows a **strictly smaller error bound** than naïve INT4 under LSB-dominant distributions, behavior PoT methods do not share.

Reviewers appreciated the clarification that QuaRTZ is a **compression mechanism** rather than a new exponent-based quantizer, and that its representational geometry, error properties, and hardware behavior are fundamentally different from PoT/FP4 formats.

### **2. Addressed concerns about GPU efficiency and implementation feasibility.**

Several reviewers were curious whether QuaRTZ could run efficiently on today’s GPUs.
We clarified with latency measurements on RTX 4090 GPU.

* Table 1, Table 7, and Table 8 report latency overhead from compression / decompression, latency of attention layers, and latency of linear layers, respectively.
* LZS compression/decompression overhead is **0.015–0.69 ms**, typically **<2–3%** of GEMM time.
* No additional global memory traffic is introduced.
* The per-group FLAG adds only a single `<<` shift to the s32 accumulator.

### **3. Added direct comparisons against MXFP4/NVFP4 and clarified accuracy behavior.**

Reviewers asked for fair comparisons against FP4 formats.
We added:

* Table 4: **direct MXFP4/NVFP4 comparisons**, confirming QuaRTZ is competitive—often superior—on FLUX.1-schnell and comparable on PixArt-$\Sigma$.
* Discussion explaining why **error accumulation differs between 4-step and 10-step models** as a key for interpreting PixArt-$\Sigma$ results.

### **4. Provided comprehensive operator-level analysis and comparisons against SVDQuant.**

Reviewers requested latency/memory comparisons against SVDQuant.
We added:

* A full breakdown of compression and decompression overhead (Table 1).
* End-to-end memory and latency comparisons (Figure 4).

This clarifies the **speed–accuracy–memory trade-off**: unlike SVDQuant, QuaRTZ is a **pure INT4 method with no FP16 branch**, making it more attractive for edge and low-power environments.

---

# **Final Note to AC**

The work introduces a **lightweight, integer-only INT4 quantization scheme** that:

* preserves accuracy across multiple diffusion models,
* delivers competitive FP4-level performance without exponent fields,
* is efficient and implementable on current GPUs,
* enjoys a clean mathematical justification,
* and offers a distinct contribution compared to PoT, MXFP4, NVFP4, or SVDQuant.

We respectfully ask the AC to consider the strength and clarity of the final submission after rebuttal.

---

### Meta-Review · Area_Chair_xSgv · 2025-12-07

**Summary:**

Three out of the four reviewers are inclined to reject the paper, and one reviewer raised further concerns during the rebuttal phase. The paper needs more writing effort. Fig. 1 describes a pipeline as key contribution, which can be done in a better way and will be more friendly for reviewers.

**Reviewer Concerns:**

7YGF concern on the quality of the paper on idea description and reference.

**Reviewer Scores:**

dbzA give higher score than other reviewers, but during the rebuttal phase more concerns are raised.
Other reviewers scores are 4 or 2, and they are not satisfied with the paper quality, such as writing problem, experimental results.

---

### Decision · Program_Chairs · 2026-01-26

Reject